# Masked Graph Neural Networks for Unsupervised Anomaly Detection in Multivariate Time Series

**DOI:** 10.3390/s23177552

**Published:** 2023-08-31

**Authors:** Kang Xu, Yuan Li, Yixuan Li, Liyan Xu, Ruiyao Li, Zhenjiang Dong

**Affiliations:** 1School of Computer Science, Nanjing University of Posts and Telecommunications, Nanjing 210003, China; kxu@njupt.edu.cn (K.X.); b20030115@njupt.edu.cn (Y.L.); b20030504@njupt.edu.cn (R.L.); 2State Key Laboratory of Smart Grid Protection and Control, Nari Group Corporation, Nanjing 211106, China; xuliyan@sgepri.sgcc.com.cn; 3School of Computer Science and Engineering, Southeast University, Nanjing 211189, China; yixuanli@seu.edu.cn

**Keywords:** unsupervised anomaly detection, multivariate time series, graph neural network, masked strategy

## Abstract

Anomaly detection has been widely used in grid operation and maintenance, machine fault detection, and so on. In these applications, the multivariate time-series data from multiple sensors with latent relationships are always high-dimensional, which makes multivariate time-series anomaly detection particularly challenging. In existing unsupervised anomaly detection methods for multivariate time series, it is difficult to capture the complex associations among multiple sensors. Graph neural networks (GNNs) can model complex relations in the form of a graph, but the observed time-series data from multiple sensors lack explicit graph structures. GNNs cannot automatically learn the complex correlations in the multivariate time-series data or make good use of the latent relationships among time-series data. In this paper, we propose a new method—masked graph neural networks for unsupervised anomaly detection (MGUAD). MGUAD can learn the structure of the unobserved causality among sensors to detect anomalies. To robustly learn the temporal context from adjacent time points of time-series data from the same sensor, MGUAD randomly masks some points of the time-series data from the sensor and reconstructs the masked time points. Similarly, to robustly learn the graph-level context from adjacent nodes or edges in the relation graph of multivariate time series, MGUAD masks some nodes or edges in the graph under the framework of a GNN. Comprehensive experiments are conducted on three public datasets. According to the experimental findings, MGUAD outperforms state-of-the-art anomaly detection methods.

## 1. Introduction

To guarantee that network systems operate normally, large amounts of industrial data are monitored at all times. The data come from numerous interrelated monitoring sensors, which are continuously generated through the operation of the system and constitute multivariate time-series data. Time-series anomaly detection [1] is a core task of intelligent operation and maintenance, and its goal is to analyze the time-series changes and find the outliers or sequences that do not match the expectations from a large number of samples, i.e., the anomalies of system hardware and software services. As the service system becomes large and complex, the use of sensors and the Internet of Things (IoT) is progressively expanding, and it is especially important to detect faults and ensure the system’s safety via monitoring.

Unsupervised anomaly detection is more widely used since the prevailing scenarios often lack stable anomaly signs and the anomaly changes irregularly. Intuitively, the task can be accomplished by setting a threshold and identifying data that exceed the threshold as anomalous. However, this approach cannot cope with the complexity and diversity of exceptions and data types. For example, anomalies may occur where the absolute value of the deviation is not very large but the trend of the data is different. Traditional unsupervised anomaly detection utilizes statistical learning methods, such as principal component analysis [2], distance-based clustering, and density-based clustering, but these methods require a priori knowledge about the anomalies. Machine learning methods, such as random forest, isolated forest [3], and one-class support vector machine [4], have also been applied to anomaly detection, but these methods are relatively simple in fitting anomalous data distributions and are not sufficient to accurately detect anomalies in multivariate time-series data.

Deep learning techniques have become popular in the area of anomaly detection by allowing neural networks to learn characteristics because of their strong learning ability and high adaptability [5,6]. For example, sequential models like recurrent neural networks (RNN) and long-short-term memory (LSTM) [7] can learn complex temporal dependencies in time series. However, their complex computation patterns lead to model performance degradation. The parallel computation of Transformers [8] achieves performance improvement but cannot exploit correlation information among multivariate sequences. Generative models, such as variational auto-encoder (VAE) [9] and generative adversarial networks (GANs) [10], reconstruct data to learn the normal data distribution and compare it with the original data to obtain the anomaly score. However, this model also does not effectively exploit the causality among different time series.

Recently, graph neural networks (GNNs) have been applied to anomaly detection [11]. GNNs are better suited for using spatial information causality between sensors for unsupervised anomaly detection because they can benefit from the internal structure information [12]. In [13], the graph’s structure was manually constructed, and the spatial information was encoded using a graph convolutional network (GCN) [14]. However, for datasets without clear graph topologies (the relationship between sensors is sometimes implicit), this approach becomes impractical. Deng et al. [15] proposed the graph deviation network (GDN), which uses graph attention (GAT) [16] to encode spatial information and the similarity between node embedding vectors to learn the graph structure among sensors. However, the GDN disregards any possible temporal dependence present within the time series and cannot learn the complex correlations among temporal samples. Also, the GDN cannot robustly model the relationships between sensors.

To address the problems with the above methods, we propose a masked graph neural network for unsupervised anomaly detection (MGUAD), a novel method that uses a GNN with masking strategies to robustly learn the temporal context from time-series data and the graph-level context from multiple time-series data (e.g., interactions between different time series) for anomaly detection. The observed time-series data from multiple sensors often lack explicit graph structures. MGUAD can model the relations among sensors as a graph and dynamically update the graph structure based on time-series data over time. A GAN framework (namely, the generator and discriminator) is used to train MGUAD. The GNNs are employed as a generator, and a discriminator is present to distinguish between the original and generated time-series sequences. To ensure that the proposed model is robust and can make the most of the learned correlations among sensors, multi-masking strategies are adopted to model the time series and graph structure. In terms of masking on time-series data, MGUAD randomly masks time points from the sequence and then reconstructs the time points through the neural network, which can adequately learn the temporal correlation of the contexts in the sequence to recover the masked part. Meanwhile, by masking the nodes or edges, MGUAD can learn the graph-level context from multiple sensors robustly within the framework of the GNN.

To summarize, our main contributions are as follows:For the purpose of unsupervised multivariate anomaly detection, we propose a novel network design that can exploit the temporal correlations of time series and the complex relationships among different sensors. MGUAD is the latest example of a GNN applied to multivariate time-series anomaly detection.We are the first to introduce the masking operation into time series and graph structures, and we use two masking strategies to enhance the learning capabilities of the model.Extensive experiments on three publicly available datasets demonstrate that our model outperforms all current state-of-the-art methods.

## 2. Related Works

The goal of anomaly detection is to identify samples that are aberrant and deviate from the typical data trend. In this section, we review anomaly detection on time-series data in the existing literature, especially unsupervised methods on multivariate time series. Our model learns the distribution of temporal data through masking and GNNs, so we also provide a summary of the related works on these two topics.

### 2.1. Unsupervised Anomaly Detection on Time Series

Numerous anomaly detection techniques have been developed as a result of the diversity of anomalous patterns, data formats, and application contexts. Taken together, the three types of efficient anomaly detection techniques are as follows: clustering-based methods (Section 2.1.1), reconstruction-based methods (Section 2.1.2), and prediction-based methods (Section 2.1.3).

#### 2.1.1. Clustering-Based Methods

The clustering-based method is primarily predicated on the idea that normal data samples are located closer to the local clustering centroid while anomaly samples are located further away. The distance between each sample and the closest clustering centroid serves as the anomaly score. Clustering-based methods mainly include the Gaussian mixture model [17], K-nearest neighbor [18], K-means [19], local outlier factor [20], etc. The one-class classification method can be considered a clustering-based method, which detects anomalies by building the decision boundary between normal and abnormal samples, such as in the one-class SVM [4] algorithm. However, clustering-based algorithms require some a priori knowledge of the anomaly information, which imposes significant limitations on the task of anomaly detection. These methods cannot effectively capture the temporal correlations of time series and have difficulty in handling high-dimensional time-series data.

#### 2.1.2. Reconstruction-Based Methods

The potential distribution of time-series data can be learned using a reconstruction-based method. These methods are based on the idea that anomalies lose information when mapped to a lower-dimensional space, making it impossible for them to be successfully reconstructed. Because of this, the anomaly score is estimated using the reconstruction loss, as seen in Principle Component Analysis (PCA) [2]. Autoencoders (AEs) are important dimensionality reduction tools in representation learning [21], with VAEs [9] commonly used in the field of anomaly detection [22,23]. However, dimensionality reduction by AEs can often lead to overfitting. VAE learns the mean and variance from the real data, whereas GANs [10] learn the distribution of the real data, resulting in better performance. AnoGAN [24] used a GAN for anomaly detection, mapping samples to latent space and reconstructing them, calculating an anomaly score through reconstruction error, but this method was intended for processing images. MAD-GAN [25] utilized a GAN for multivariate anomaly detection, capturing temporal correlations from time series by using LSTMs as the generator and discriminator. For anomaly detection of heartbeat signals, BeatGAN [26] utilized a combination of an autoencoder and a generative adversarial network. Although the GAN model has better learning capabilities, GAN-based methods can hardly account for the complex correlations among time sequences. GANs require the use of appropriate foundational models to consider correlation and temporal dependence.

#### 2.1.3. Prediction-Based Methods

Prediction-based methods learn to fit the time series and then predict the values at the next moment. If there is a significant disparity between the predicted data and the initial sample (e.g., the three-sigma criterion [27]), this data point will be identified as an anomaly. Prediction-based methods do not learn the distribution of the anomalous data directly, but they are able to learn the original data distribution. The new unseen anomalies will be different from the original data, so these methods can use this difference to determine the occurrence of anomalies and can handle unseen anomalies. Traditional statistical models like ARIMA and Holt–Winters [28] can fit time-series data, but these models generally require strong assumptions about the data distribution. Classical deep learning methods such as RNN and LSTM [7] can better capture the complex relationships within time series, but the long-term dependence of serial computation can lead to the degradation of model performance, while the learning mode of recursive operations is inefficient. The Transformer [29]-based method can be used to learn the context dependence of time series. Additionally, its multi-attention mechanism allows for parallel operations, effectively addressing the shortcomings of recurrent neural networks. But Transformers cannot exploit the dependencies between different time series. Recently, graph neural networks have been applied to multivariate time-series data. The graph neural network assumes that each node is affected by its neighbors, thus enabling the correlations among nodes to be effectively used to model the graph-structured data. A GDN [15] treats each time series as a node in a graph and uses the attention-based method to forecast future values.

### 2.2. Masking-Based Methods

The masking operation, a common method used to improve a model’s learning capabilities, has received widespread application across several deep learning tasks. In general, the masking operation removes or replaces a portion of the input to the neural network, which can improve the model’s capacity by reconstructing the masked data. In the natural language processing field, BERT [30] employs the masked language modeling task for language representation learning. The model is pre-trained by predicting the words in the blocked or replaced sentences. In computer vision tasks, certain portions of the image are randomly masked, and then the model reconstructs the masked pixels [31]. This proxy task is similar to the masking prediction in BERT but differs in that it predicts all the pixels within the block. In terms of time series, ref. [32] explores the interpretability of time-series prediction using dynamic masking methods. Ref. [33] performs temporal generation using a masked autoencoder. Ref. [34] improves the learning capabilities of the model by estimating time series through random masking for self-supervised learning. This study is the first we are aware of to apply masking methods for time-series anomaly detection.

### 2.3. Graph Neural Networks

Large amounts of data in the real world, such as social networks, knowledge graphs, complex file systems, etc., are unstructured. The emergence of GNNs addresses the limitation of traditional neural networks, which are only effective in processing structured data such as sequences and grids. GNNs are good at modeling intricate patterns in data using a graph structure. In general, the key design element of GNNs is that graph nodes exchange information with their neighbors to update their representations. Graph convolution networks (GCNs) [14,35] use convolutional operations similar to those in image processing and provide a concrete derivation for this model type. Instead of focusing on the entire dataset, graph attention networks (GATs) [16] direct attention to the important portion of the data and introduce the self-attention mechanism, which assigns different weights to each node in the graph based on its different characteristics. Various variants of GNN-based models have also been applied to time-dependent tasks such as traffic prediction [36] and time-series forecasting [37]. However, GNNs require graph-structured data as input, which, in our case, are frequently unknown and must be captured from the data. STGs [13] manually construct graph-structured data and use GCNs to encode spatial information. For datasets lacking clear graph topologies, this method can become impractical. GDNs [15] treat each time series as a node in a graph, learn the graph structure by calculating the node similarity, and use a GAT to calculate each node’s anomaly score. However, GDNs do not consider the temporal correlation within the time series and cannot learn the complex correlations within temporal samples. Moreover, these methods cannot robustly model the relationships between sensors.

## 3. Methodology

### 3.1. Overview

Our research attempts to address the issue of multivariate time-series anomaly detection. A multivariate time series is represented by $X={\left\{x_{t}\right\}}_{t\in T}$, where $x_{t}=\left(x_{t}^{1},\ldots ,x_{t}^{n}\right)$, which includes *n* variables (sensors), and $t\in T\subseteq Z^{+}$ indicates a specific time point. The anomalies often occur in a time point or slice, so the anomaly data are defined as $A=\left\{x_{p},x_{p+1},\ldots ,x_{p+r-1}\right\}$, where *p* is the anomaly’s starting time point, and $r\in \left(1,\left|T\right|\right)$ is the anomaly’s duration. In short, our aim is to detect all the anomalies in the observed data.

Figure 1 illustrates the architecture of the MGUAD model. During the training process (black line), we take multivariate time series from numerous sensors as the inputs and model relationships between sensors as a graph. Then, MGUAD uses the masks to remove a portion of each time series, adopts a GNN to generate time-series data, and minimizes the difference between the reconstructed and the input data through a supervised loss (Equation (Equation 7)). The model reconstructs the masked data, thereby obtaining a learned distribution of normal data. Then, MGUAD randomly masks the edges of the graph, uses a graph attention function over its neighbors to forecast the future values of each sensor, and employs a discriminator to determine whether the predicted value is real. During testing, we take time-series data as input, use the trained model to predict future values, and compute the deviations as the anomaly scores between the predicted behavior and the observed behavior. Finally, the anomaly scores can be used as judgments for anomaly detection.

### 3.2. Graph-Structure Learning

In this section, we describe how to capture the relationships between multivariate time series and build a graph to utilize the correlations between sensors. The graph’s nodes correspond to individual sensors. Let G=(V,E) be a directed graph, where *V* is a finite set of nodes and *E* is a finite set of edges. We use $s\left(e\right)=\left(v_{1},v_{2}\right)$ to denote that the edge e∈E connects $v_{1}\in V$ and $v_{2}\in V$, which is an edge from a to b. The graph edge between two nodes indicates that there is a dependency relationship between them. Therefore, $s\left(e\right)=\left(v_{1},v_{2}\right)$ indicates that $v_{1}$ has an influence on the monitoring data of $v_{2}$. Since the relationships are not necessarily mutual, directed instead of undirected graphs are used in MGUAD. The graph structure is initialized before the training of each application and when the application scenario changes, resulting in changes in the sensor relationships. The graph model undergoes automatic updates by adjusting the node embeddings and exploiting the similarity of the embeddings during the training process.

#### 3.2.1. Graph-Structure Construction

To build a graph structure, we need to obtain the complex relationships between the sensors. The similarity of data in time series can be used to calculate the correlation between two sensors. The first *m* samples of each time series are selected to represent the initial behavior of these sensors, and we calculate the Pearson correlation coefficient between them to obtain the correlation between two time series. The values of the similarity are normalized and are used as the weights of the edges between nodes. To control the scale of the graph, which can eliminate unnecessary edges and improve the efficiency of the operation, we use two strategies to control the number of edges: filtering out edges between nodes with a similarity lower than *s* and constraining the maximum number of neighbors per node to *n*.

We select the data within the first *m* time points in each time series to calculate the initial correlations between time series. $x^{k}=\left(x_{t}^{k},\ldots ,x_{t+m-1}^{k}\right)$ is the *k*th time series with an *m* number of timestamps. We calculate the Pearson correlation coefficient between two series to obtain their similarity. As shown in the formula below, $E(x^{k})$ denotes the mean of the series, $\sigma_{x^{k}}$ denotes the standard deviation of the series, and $Cov(x^{k1},x^{k2})$ represents the covariance between the two series.
(1)$$E\left(x^{k}\right)=\frac{\sum_{i=t}^{m+t-1}x_{i}^{k}}{m},\sigma_{x^{k}}=\sqrt{\frac{\sum_{i=t}^{m+t-1}{\left(x_{i}^{k}-E\left(x^{k}\right)\right)}^{2}}{m}},Cov\left(x^{k1},x^{k2}\right)=\frac{\sum_{i=t}^{m+t-1}\left(x_{i}^{k1}-E\left(x^{k1}\right)\right)\left(x_{i}^{k2}-E\left(x^{k2}\right)\right)}{m}$$

The relationship $R_{t}\left(v_{k1},v_{k2}\right)$ between two sensors can be calculated as follows: (2)$$R_{t}\left(v_{k1},v_{k2}\right)=\frac{Cov(x^{k1},x^{k2})}{\sigma_{x^{k1}}\sigma_{x^{k2}}}=\frac{\sum_{i=t}^{m+t-1}\frac{\left(x_{i}^{k1}-E\left(x^{k1}\right)\right)}{\sigma_{x^{k1}}}\frac{\left(x_{i}^{k2}-E\left(x^{k2}\right)\right)}{\sigma_{x^{k2}}}}{m}$$
We can obtain the similarity between one node and the others and then select the largest *n* among them as candidate neighbors *i*: $N_{c}\left(i\right)=\left\{v_{1},v_{2},\ldots ,v_{n}\right\}$. $E_{c}\left(i\right)=\{R_{t}\left(v_{i},v_{1}\right)$, $R_{t}\left(v_{i},v_{2}\right),\ldots ,$$R_{t}\left(v_{i},v_{n}\right)\}$ is the set of correlation values between *i* and each node in $N_{c}\left(i\right)$. We normalize $R_{t}$ to be the weights of the edges $e_{i,j}$ for graph attention-based prediction as follows: (3)$$e_{i,j}=Norm\left(R_{t}\left(v_{i},v_{j}\right)\right)=\frac{R_{t}\left(v_{i},v_{j}\right)-min\left(E_{c}\left(i\right)\right))}{max\left(E_{c}\left(i\right)\right)-min\left(E_{c}\left(i\right)\right)}$$

Then, we eliminate the edges with weights lower than *s* to streamline the graph structure. Next, we obtain the initial composition of the graph structure.

#### 3.2.2. Graph-Structure Updating

Relationships between sensors can change over time (e.g., the closing of flow control valves can lead to relationships transforming in a local scope, as the water flow is suddenly reduced and cannot affect other sensors anymore), and MGUAD should update the graph structure as the state changes. For graph-structure learning, we map each sequence as a node embedding (high-dimensional vector). We use node embedding to represent each sensor’s features. The node embedding vectors are initialized randomly and are updated with the model training. We represent embeddings as $V_{i}\in R^{d}$, and the similarity between them indicates their behavioral similarity, which can be used in graph-structure updates.

By calculating the cosine similarity between node embeddings, we can obtain the correlations between node features using $R_{n}\left(v_{1},v_{2}\right)$, where $V_{1}$ and $V_{2}$ are the embeddings corresponding to nodes $v_{1}$ and $v_{2}$.
(4)$$R_{n}\left(v_{1},v_{2}\right)=\frac{V_{1}\cdot V_{2}}{∥V_{1}∥∥V_{2}∥}$$

Graph-structure learning also needs *m* samples of the time series from the current time period. Similar to how we proceed during the initial graph creation, these samples represent the behaviors of the sensors during this time period. MGUAD uses both node embedding and sampling in tandem for graph-structure learning. We calculate $R_{t}\left(v_{1},v_{2}\right)$ using Equation 2 based on the current *m* samples and normalize and assign weights to the sum to obtain the new similarity. We then update the graph structure in a similar way to graph building as follows: (5)$$e_{i,j}=\gamma_{1}Norm\left(R_{t}\left(v_{i},v_{j}\right)\right)+\gamma_{2}Norm\left(R_{n}\left(v_{i},v_{j}\right)\right)$$
where $\gamma_{1}$ and $\gamma_{2}$ are two adjustable weights, and their relative size determines whether the sensor relationships focus on sample values or node embeddings in MGUAD.

### 3.3. Masked Temporal Feature Reconstruction

Regarding reconstructive learning, by randomly masking multivariate sequences, MGUAD can be forced to reconstruct the sequences by leveraging the context of each sequence and the correlations between different sequences, thereby enhancing the model’s ability to handle complex data. To learn the time correlations in time series, we initially remove a portion of the input time series and then instruct the network to recreate the original input time series. We remove a random portion of the sequence determined by the sliding window. For example, when using a 20% masking ratio for a sliding window with a length of 20, we randomly mask four time points. Our goal is to learn contextual information within time series, so to break the continuity in the time series on a large scale, we specify the maximum sequence length for the masked subsequences. Generally, the maximum continuous masking length will not exceed half of the masking ratio. Thus, the longest consecutive masked time points for a 20% masking ratio of 20-length inputs would be 2.

To reconstruct the masked time series, the model can learn the temporal dependencies between different time points. Specifically, MGUAD generates values at the masked time points and calculates the reconstruction loss by comparing these values to the original data before masking. For example, a sequence $x^{k}=\left(x_{t}^{k},\ldots ,x_{i}^{k},\ldots ,x_{t+w-1}^{k}\right)\in X^{t}$ input with a length of w will be randomly masked and become $\left[x^{k}\right]=\left(x_{t}^{k},\ldots ,\left[\mathrm{mask}\right],\ldots ,x_{t+w-1}^{k}\right)\in \left[X^{t}\right]$, where G(…) represents the graph structure model, and the $x_{i}^{k}$ is reconstructed using G(…).
(6)$$G\left(\left[X^{t}\right]\right)⟹\left(x_{t}^{k},\ldots ,\hat{x_{i}^{k}},\ldots ,x_{t+w-1}^{k}\right)$$

The reconstruction loss, $loss_{rec}$, can be calculated as
(7)$$loss_{rec}=\frac{1}{n\times \left|\mathrm{mask}\right|}\sum_{i=1}^{n}\sum_{j=1}^{\left|\mathrm{mask}\right|}∥\hat{x_{j}^{i}}-x_{j}^{i}∥,x_{j}\in \left[mask\right]$$

G(…) is optimized by minimizing $loss_{rec}$, which is carried out through joint training with the entire model.

### 3.4. Masked Graph Predictions

This section includes instructions for using graph structures to formulate predictions. To enhance the robustness of the model, we mask the nodes or edges of the graph. During prediction, the model can only use incomplete data after applying a random mask as input, which enhances its capability to extract complex information from within and between sequences, ultimately improving its ability to leverage unbalanced or ambiguous data. The idea of using graph structures to calculate predicted values through an attention mechanism comes from GAT [16], where we aggregate information about a node and its correlation with its neighbors. We can apply node embeddings to the graph attention mechanism, which permits heterogeneous effects for various kinds of sensors. The attention coefficients $\alpha_{i,j}$ between graph nodes need to be calculated. The value of $\alpha_{i,j}$ demonstrates how significant node j’s features are to those of node i. We denote the neighbors of node *i* as N(i). To calculate $\alpha_{i,j}$, we consider the influence of two nodes simultaneously and calculate the attention value $\epsilon_{i,j}$ between them: (8)$$\epsilon_{i,j}=LeakyReLU\left(a\left(\left(Wx_{t}^{i}⊕V_{i}\right)⊕\left(Wx_{t}^{j}⊕V_{j}\right)\right)\right)$$
where LeakyReLU is a nonlinear activation function, *a* is a vector of learning coefficients for the attention mechanism, ⊕ denotes concatenation, and W is a trainable linear transformation matrix with shared weights at each node. By combining the node embedding $V_{i}$ with the current moment’s feature $x_{t}^{i}$, a comprehensive attention factor can be created. When aggregating neighbor information, the attention of all neighbors needs to be normalized. The attention weights after normalization are the aggregation coefficients $\alpha_{i,j}$, which are calculated as follows: (9)$$\alpha_{i,j}={\mathrm{softmax}}_{j}\left(\epsilon_{ij}\right)=\frac{exp(\epsilon_{ij})}{\sum_{k\in N(i)}exp\left(\epsilon_{ik}\right)}$$

We obtain the aggregated representation $\chi_{t}^{i}$ of each node using the aggregation coefficients $\alpha_{i,j}$ as follows: (10)$$\chi_{t}^{i}=LeakyReLU\left(\alpha_{i,i}Wx_{t}^{i}+\sum_{j\in N(i)}\alpha_{i,j}Wx_{t}^{j}\right)$$

Then, we input the aggregated representation $\chi_{t}^{i}$ of all nodes into a multi-layer fully connected network M(…) to obtain the predicted values for each node. We denote this whole prediction process as G(…): (11)$${\hat{X}}^{t+1}=G\left(X^{t}\right)=M\left(\chi_{t}^{1},\chi_{t}^{2},\ldots ,\chi_{t}^{n}\right)$$

Similar to the dropout operation [38] in deep learning, MGUAD randomly removes nodes and their connected edges before prediction to improve the robustness of the model. This means that the input of the graph attention network is a masked graph from which some relationships between sensors have been removed. To this end, we employ two graph-masking strategies: one involves excluding a random portion of nodes when constructing and updating the graph, whereas the other involves setting the weights of some edges to 0 during training, preventing them from participating in graph attention operations. Experimentally, we found the latter approach to be more effective, as it allowed us to control the retention of edges with higher weights, thus preventing the model from losing important inter-series correlation information. We mask the edges during training to ensure that the model comprehends the full relevance information during inference. Through forecasting, we obtain a model that has enough power to predict future values.

### 3.5. Adversarial Training

The main goal of prediction-based anomaly detection methods is to develop a model that can accurately anticipate future values. Anomaly scores are computed by the difference between the true value and the predicted value. The deviation between the expected and actual behavior of the time series is the key to anomaly detection. MGUAD should model the behavior of time series, so MGUAD needs to learn to predict the future value close to the true value. We apply an adversarial learning approach to learn the “normal” behavior of time series. In such a structure, the two players in a two-player min-max game are the generator and the discriminator. The generator attempts to produce samples that may deceive the discriminator, whereas the discriminator attempts to discern between real samples and generated ones.

Specifically, when training in an adversarial manner, we need to optimize the generator to fool the discriminator. The input obtained through the time window is represented by $\left\{x_{1},\ldots ,x_{t-1}\right\}$. We can obtain $x_{t}$ using the graph-based model G(…). The loss of the generator, $loss_{G}$, can be calculated using Equation (Equation 12): (12)$$loss_{G}=\frac{1}{n}\sum_{i=1}^{n}\log\left(-D\left(G\left(x_{1}^{i},\ldots ,x_{t-1}^{i}\right)\right)\right)$$

We use a Transformer as a discriminator, denoted as D(…). Transformers are effective in processing time-series data [39,40]. They are also more efficient compared to traditional models like RNNs because of their parallel calculation capability. MGUAD uses a Transformer as a discriminator, which must differentiate between the time-series sequences generated using G(…) and the real time-series sequences. The input of the discriminator is two time series: one is a normal time series, except for the last time point, which contains the predicted values using G(…), and the other is the normal data. Specifically, the discriminator can distinguish between two time series that only differ at the last time point, one of which is the true series and the other containing a predicted value. We denote this discriminator as $D\left(\ldots \right)$. Let $x_{t}$ be the real data and $\hat{x_{t}}$ be the prediction. The input can be presented as $x_{t}\Rightarrow D\left(x_{1},\ldots ,x_{t}\right),\hat{x_{t}}\Rightarrow D\left(x_{1},\ldots ,\hat{x_{t}}\right)$. Then, the discriminatory loss, $loss_{d}$, can be calculated as
(13)$$loss_{D}=\frac{1}{n}\sum_{i=1}^{n}\left[-\log D\left(x_{1}^{i},\ldots ,x_{t}^{i}\right)-\log\left(1-D\left(x_{1}^{i},\ldots ,\hat{x_{t}^{i}}\right)\right)\right]$$

Ultimately, we sum three loss functions, the reconstruction loss, $loss_{rec}$, which represents the reconstruction accuracy of the masked timestamp, the generation loss, $loss_{G}$, which reflects the ability of the generator to predict future values, and the discriminant loss, $loss_{D}$, which is used to distinguish between the real and generated time series. We train the entire model uniformly by integrating the three loss functions using weights: (14)$$loss_{total}=\lambda_{1}loss_{rec}+\lambda_{2}loss_{G}+\lambda_{3}loss_{D}$$
where $\lambda_{i},i\in 1,2,3$ controls the relative importance of the three terms. $loss_{G}$ and $loss_{D}$ undergo an asynchronous training process, which is demonstrated in Algorithm 1.

### 3.6. Anomaly Score

In the inference phase, we calculate an anomaly score for each time series. A higher score means the time series is more likely to be anomalous. The anomaly scores consist of two components: deviation scores from the deviation between the predicted value and the true value, and critic scores from the discriminator. Intuitively, the deviation is a measurement that indicates the abnormal behavior of the time series.

To reduce the effect of the relative sizes of the different series, we use smoothing scores when calculating the anomaly scores, i.e., by subtracting the mean of each series and dividing by the standard deviation.
(15)$$dev_{n}=\frac{\left|x_{t}^{n}-\hat{x_{t}^{n}}\right|-\mu }{\sigma }$$
where μ and σ are the mean and standard deviation of each series.

The largest normalized anomaly score of all the sequences can be selected as the anomaly score $ADS_{t}$ for moment *t*.
(16)$$ADS_{t}=\max{\left\{loss_{n}\right\}}_{n\in N}$$

We consider moments where the anomaly score exceeds a threshold as anomalous time points, and nodes with large anomaly scores as anomaly sensors.

The overall algorithm is shown in Algorithm 1.
**Algorithm 1** Multivariate Time-Series Anomaly Detection Algorithm**Training:****Input:** $X^{Train}=\left\{x_{1},\ldots x_{t}\;\right\}\subseteq$ Train dataset;**Output:** model parameters such as generator parameters: G()**if** epochs within the number of training iterations **then**   **for** each epoch **do**     node embedding: $N=\left(n_{1},\ldots ,n_{m}\right)\Rightarrow E\left(N\right)$     Node masking, drop graph nodes by a certain percentage     **for** each nodes n∈N **do**        Calculate the embedding similarity with other nodes        Fill in the adjacency matrix with the Top-k nodes with the highest similarity as neighbors     **end for**     Time-series masking: $X=\left\{x_{1},\ldots x_{t-1}\;\right\}\Rightarrow M\left(X\right)$     Calculate the attention scores by the adjacency matrix     Reconstruct the masked data through the attention mechanism: $M\left(x_{1},\ldots ,x_{t-1}\right)\Rightarrow G\left(X\right)=\hat{X}$     Predict the next moment *t* through the attention mechanism: $X\left(x_{1},\ldots ,x_{t-1}\right)\Rightarrow G\left(X\right)=\hat{x_{t}}$     Discrimination: $x_{t}\Rightarrow D\left(x_{1},\ldots ,x_{t}\right),\hat{x_{t}}\Rightarrow D\left(x_{1},\ldots ,\hat{x_{t}}\right)$     Update discriminator: $\min\frac{1}{n}\sum_{i=1}^{n}\left[-\log D\left(x_{1}^{i},\ldots ,x_{t}^{i}\right)-\log\left(1-D\left(x_{1}^{i},\ldots ,\hat{x_{t}^{i}}\right)\right)\right]$     Update generator: $\min\sum_{i=1}^{n}\log\left(-D\left(G\left(x_{1}^{i},\ldots ,x_{t-1}^{i}\right)\right)\right)+\sum_{i=1}^{n}\sum_{j=1}^{\left|\mathrm{mask}\right|}∥\hat{x_{j}^{i}}-x_{j}^{i}∥,x_{j}\in \left[mask\right]$     Record parameters in the current iteration   **end for****end if****Inference:****Input:** $X^{Test}=\left\{x_{1},\ldots x_{t}\;\right\}\subseteq$ Test dataset;**Output:** Time points when anomalies occurred: $T_{a}=\left\{t_{a1},t_{a2},\ldots ,t_{an}\right\}$Predicting the value: $X^{Test}=\left\{x_{1},\ldots x_{t-1}\;\right\}\Rightarrow G\left(X^{Test}\right)=\hat{X}$**for** each nodes n∈N **do**   Calculate the loss between the predicted value and the true value for each node: $loss_{n}=\frac{\left|x_{t}^{n}-\hat{x_{t}^{n}}\right|-\mu }{\sigma }$**end for**time *t*’s anomaly detection score: $ADS_{t}=\max{\left\{loss_{n}\right\}}_{n\in N}$Get anomaly time points $T_{a}=\left\{t_{a1},t_{a2},\ldots ,t_{an}\right\}$ by $ADS_{t}$

## 4. Experiments

We first describe the utilized datasets and experimental settings in this section. Then, the results of the experiments are shown, and we subsequently analyze them.

### 4.1. Datasets

Three public datasets are utilized in our experiments to evaluate our model, as shown in Table 1: SWaT [41] (https://mlad.kaspersky.com/swat-testbed/, accessed on 10 August 2022), WADI [42] (https://itrust.sutd.edu.sg/itrust-labs_datasets/dataset_info/, accessed on 10 August 2022), and KDDCUP99 (http://kdd.ics.uci.edu/databases/kddcup99/kddcup99.html, accessed on 10 August 2022). They are all multivariate time-series datasets and contain sufficient data for training and testing.

The SWaT dataset is related to attack tests on water distribution systems and water security treatment systems conducted at the Cyber Security Centre of the Singapore University of Technology and Design. This dataset represents tests for water purification used in cyber security research. The SWaT dataset comprises 264 h of numerical and network traffic data that were gathered over the course of 11 consecutive days by 51 sensors and processors. It comprises 4 days of abnormal data collected when the system was under assault under various conditions, as well as 7 days of normal data collected when the system was functioning as normal. Specific malware attacks include historian data exfiltration attacks and process disruption attacks. These attacks may result in a change in the sensor detection data, a sudden pause in normal supply, etc.

The WADI and SWaT datasets originate from the same laboratory. The KPIs of the WADI dataset are derived from the data collected by 123 sensors and actuators in a water distribution system. This distribution system, which consists of a large number of water distribution pipes, is more vulnerable and, therefore, the WADI dataset contains more features compared to the SWaT dataset. Data were collected from the network, sensors, and actuators for 16 consecutive days, including 14 days of normal operation and 2 days of abnormal operation. Fifteen attacks using the same attack model are contained in the anomaly data. Due to the anomaly rate being lower compared to the other datasets, the WADI dataset is more unbalanced.

The KDDCUP99 dataset was derived from competition data from the 1999 KDD CUP competition, which simulated the US Air Force LAN environment and monitored 34 categories of key performance indicators. The competition’s main requirement was to create a network intrusion detector that could determine whether a network connection was being attacked or experiencing intrusion. Subsequently, each network connection was classified as either “attack” or “normal”. The anomalies were the result of simulated attacks on the local area network. These attacks encompassed 39 different anomaly types, 22 of which appeared in the training set and the remaining 17 appeared exclusively in the test set.

These three datasets share a common characteristic, i.e., there is a close relationship among various time series, which can help demonstrate the performance of MGUAD.

### 4.2. Baselines

Nine baselines were applied in this experiment, including a statistics method (principal component analysis (PCA)), a machine learning method (isolation forest), and a traditional deep learning method (LSTM). The remaining baselines were complicated combined models, which have proven to achieve outstanding performance in related tasks in recent years.

**PCA**: Principal component analysis [2] is a linear dimensionality reduction method that projects time-series data in different directions, reflecting the difference in the variance of the original data and the intrinsic characteristics of the series variation. PCA judges data samples that deviate greatly from other data samples in certain directions as outliers.**Isolation Forest**: Isolation forest [3] is an efficient machine learning anomaly detection algorithm, which divides data points in a time series into a tree and clusters data with the same properties into one class. It uses the position of the divided nodes in the tree for anomaly determination. The lower the depth of a node, the easier the data division, which implies their status as outliers.**LSTM**: LSTM [7] is a classical deep learning model that mines the contextual relationships in time series through recursive operations. The anomaly detection algorithm based on LSTM mainly works through the method of prediction, i.e., using LSTM to learn the complex patterns of the sequence to predict the value of the sequence at the next moment. Samples with large deviations from the true value are judged as anomalous.**LSTM_VAE**: VAE is a common deep learning framework that learns the representation of data after dimensionality reduction by fitting the distribution of the data. This self-encoding architecture mainly performs anomaly detection through reconstruction, that is, it treats anomalous data as noise, considering that the data are compressed to retain only normal information while losing anomalous information. It discriminates anomalies by comparing reconstructed data with the original data. LSTM_VAE [22] combines LSTM with VAE, using LSTM to replace the feedforward network in VAE.**DAGMM**: DAGMM [17] combines an autoencoder with a Gaussian mixture model simultaneously optimizing the parameters of the deep autoencoder and the mixture model in an end-to-end manner. DAGMM uses the deep autoencoder to downscale the input data points to a low-dimensional representation while obtaining the reconstruction error. The low-dimensional representation of the data is fed into the Gaussian mixture model, which then combines the reconstruction loss to decide whether the data are anomalous.**MadGAN**: MadGAN [25] is a reconstruction-based anomaly detection algorithm. During training, to create false data, the generator receives input from the training data and random hidden variables, and the discriminator’s results are used to update the model parameters. When testing, the hidden variables that best match the distribution of the test data are first learned, and then the hidden variables are used to reconstruct the data, calculate the difference between the reconstructed sequence and the actual sequence, and finally combine them with the discriminator’s results to obtain the anomaly score.**USAD**: USAD [43] is an unsupervised anomaly detection approach for multivariate time series based on a GAN-inspired self-encoder architecture. Adversarial training enables USAD’s encoder–decoder design to learn how to increase the reconstruction error of inputs containing anomalies, thereby achieving more stability compared to conventional GANs architecture.**GDN**: GDN [15] is a prediction-based multivariate temporal anomaly detection method that treats each feature dimension of the data as a node in a graph neural network, learns the graph structure through node similarity, and calculates the anomaly score of each node using a graph attention mechanism. Finally, the method combines the anomaly scores of all temporal sequences and determines whether the moment is anomalous according to a threshold.**CAE**: CAE [23] is a diversity-driven, convolutional ensemble that combines several convolutional sequence-to-sequence autoencoder-based fundamental outlier detection models. CAE also uses a novel diversity-driven training method to maintain diversity among the base models, thus improving accuracy. This method allows for a high degree of parallelism in training and has improved efficiency.

### 4.3. Experimental Settings

Empirically, we set the size of the sliding window to 5, and the embedding dimension of the nodes to 64 (SWaT, KDDCUP99) and 128 (WADI). We trained our model using the Adam optimizer. The learning rates of the generator and discriminator were set to 0.001 and 0.0001, respectively. The maximum number of neighbors of a graph node *n* was set to 15 (SWaT, WADI) and 20 (KDDCUP99). The time-series masking ratio was set to 20%. The weights of the edge values when building the graph were set to $\gamma_{1}=\gamma_{2}=0.5$. The ratio of the weights between the three losses $\lambda_{1}:\lambda_{2}:\lambda_{3}$ during training was 1:2:1. We trained the models for up to 100 epochs and used early stopping with a patience of 10. MGUAD and its variants were implemented on a Tesla T4 graphics card. We ran the model and the nine baselines on three datasets to determine the model with the best performance.

### 4.4. Quantitative Analysis

To evaluate the anomaly detection performance, three metrics were used in our work: Precision, Recall, and F1_score. They are defined below. TP indicates the number of positive samples correctly classified, i.e., the anomaly samples were labeled as anomalies. Conversely, FP indicates the number of normal samples misclassified as anomalies. Similarly, TN is the number of normal samples correctly recognized as normal, whereas FN is the number of abnormal samples incorrectly identified as normal.
(17)$$Precision=\frac{TP}{TP+FP},Recall=\frac{TP}{TP+FN},F_{1}=2\times \frac{Precision\times Recall}{Precision+Recall}$$

Precision reflects the percentage of projected positive cases that are actual positive examples. That is, it indicates how many detected anomalies are true anomalies. Recall represents the percentage of true positive samples from the test set that the classifier has selected. In simple terms, this is the proportion of anomalies we detect out of the total number of anomalies in the sample. The Precision and Recall are diagnostic instruments for binary classification models. The F1_score is the average of Precision and Recall. Considering only one of Precision or Recall is not sufficient to evaluate the overall performance of a model; thus, the F1_score is used to take both into account as a comprehensive evaluation metric for a model. The values of these three metrics shown in Table 2 display the model’s performance with the optimal selection of parameters. The highest value of the metric in each dataset is bolded in the table.

In Table 2, we compare the anomaly detection results of the proposed model to those of the baselines for the three datasets. The results show that our model achieved the best Precision and F1_score values across all three datasets. On the SWaT dataset, MGUAD performed the best in all three metrics. Precision reached a value of 0.9851, and the combined F1_score metric improved by more than 6% compared to the second-best baseline, USAD. On the WADI dataset, although the Recall values were lower than the two baselines, they were achieved by sacrificing Precision, and our model achieved a Precision value well above the second-best baseline, with the F1 value 6.22% higher than the second-best baseline, GDN. The WADI dataset is unbalanced and has higher dimensionality than the other datasets. MGUAD performed as well on this dataset as it did on the other datasets. On the first two datasets, the deep learning method outperformed the traditional method, whereas on the KDDCUP99 dataset, the isolation forest model still exhibited good performance, indicating that the anomalous performance of the KDDCUP99 dataset is relatively typical. On this dataset, our model maintained the best performance, with all three metrics exceeding 0.96. Our model performed well on both the WADI dataset, which has highly dispersed features, and the KDDCUP99 dataset, which has only 34 features, indicating that the proposed model can adapt well to different feature dimensions. Table 3 shows the overall performance of MGUAD on the three datasets. The highest value of the metric in each dataset is bolded in the table. In general, the overall performance of our model was better compared to all the baselines.

### 4.5. Qualitative Analysis

The correlations among nodes support the construction of the graph. To analyze the effectiveness of the generated graph structure, we first analyzed the variation trend among the sensors and the similarity of the embeddings and then verified whether the connection relationships of the graph were reasonable. We selected two typical sensors: 1_MV_001, which represents the state of an electronically controlled valve, and the flow rate detected by 1_FIT_001_PV, which is controlled by this valve. As shown in Figure 2a, the two nodes exhibited the same transformation moments and trends, and there was a strong correlation between the two sensors. Figure 3a shows the graph structures associated with the sensors, named 1_FIT_001_PV and 1_MV_001_STATUS, and the PCA projection of the node embeddings. As shown in Figure 3a, the projection distance between the two sensors is close, which indicates that embeddings can be used as node features to depict dependencies between nodes.

Part of the graph structure learned by the model can be seen in Figure 3b, where the two sensors are directly connected. The weights of the edges are darker in color, indicating that the graph structure adequately captures the dependency information between these nodes.

The yellow part in Figure 2a,b shows the time interval when the anomaly occurred. Meanwhile, the electric valve 1_MV_001 was maliciously opened. Although these two sensors did not directly show a large numerical change, the other affected sensors showed abnormal monitoring values. As shown in Figure 3a,b, all three sensors—TOTAL_CONS_ REQUIRED_FLOW, 2_FIC_401_CO, and 2_LT_001_PV—are closely related to 1_MV_001 and exhibit significant anomalies in Figure 2b. Our model determined anomalies based on the anomaly scores among individual sensors, and although no anomalies were detected for the two sensors in Figure 2a, their monitoring values fluctuated for the sensors connected to the electronically controlled valve that was affected by anomalies, as shown in Figure 2b. Our model, combined with the general performance of the sensors, adequately captured the relationship between the sensors, enabling the identification of the interval where the abnormal occurrence took place. In summary, it can be seen that the graph structure can make full use of the correlation between nodes to detect anomalies.

### 4.6. Ablation Study

To assess the plausibility of the model architecture and the necessity of the masking operation, it was necessary to carry out ablation experiments. In our model, we aimed to verify whether the masking operation leads to performance improvement and whether multi-masking yields better results than individual masking. For this purpose, three approaches were used for comparison:Without masking: Retaining the original model without any masking operations implies that the model does not reconstruct the masked data and can utilize all nodes in the generated graph for predictions.With graph masking: Stochastically adding masks to the base model for graph nodes. This means that the model can only use some of their neighbors for predictions. The graph structure is not fully accessible, enhancing the model’s ability to utilize correlations among different time series.With time-series masking: Masking a portion of the input of the model, obtained through sliding windows, and then having the model reconstruct the masked samples using the unmasked generated graph structure.With both edge masking and time-series masking: The model’s predictions are obtained by combining the two masking strategies with a fine-tuned masking ratio. This operation can be called multi-masking.

To fix the other hyperparameters during the experiment, we set the masking ratio at each step to 20%. Since fixed hyperparameters were used to maintain the consistency of the experiments, the results may be slightly lower than the optimal hyperparameter settings. However, they are sufficient for determining the performance trend of the model under different masking strategies.

Table 4 shows the Precision, Recall, and F1_score metrics, reflecting the effects of anomaly detection under different masking strategies. The highest value of the metric in each dataset is bolded in the table. Figure 4 visualizes the performance improvements resulting from the masking operation, where a, b, c, and d represent the model without masking, time-series masking, node masking, and multi-masking, respectively. The red column in the diagram represents the F1_score, and it can be seen that the three masking operations improved the overall performance of the model compared to the model without the masking operation. When using the same masking ratio in all our experiments, we can see that the multi-masking operation led to significant improvements in the model’s performance across all metrics and had the most substantial impact among all the masking strategies.

We analyzed the specific characteristics of the different masking strategies, and the results are shown in Table 4. The bolded data indicate the best values for each indicator on each dataset. First, in terms of time-series masking, it can be seen that it improved the performance of the model mainly in terms of Precision, based on the superiority of the F1_score compared to the model without the masking operation. In particular, on the WADI dataset, time-series masking achieved the highest Precision value among all the masking strategies.

The use of graph masking led to non-negligible improvements in the Recall metric. Similar to time-series masking, the F1_score achieved through graph masking was also considerably improved on the WADI dataset. As can be seen, for the WADI dataset with sparse data and high dimensionality, the large improvements resulting from the use of both masking operations demonstrate their excellent capability in handling complex datasets. The Recall metric for time-series masking exhibited a slight reduction on the SWaT and KDDCUP99 datasets compared to the model without a masking operation, as did the Precision metric for graph masking. However, the indicators for multi-masking are not available due to the model’s lack of masking.

As shown in the table, the combined performance of multi-masking outperformed that of individual masking, demonstrating that multi-masking can combine the advantages of two individual masking operations and can be adapted to different datasets, making full use of the respective advantages of different masking strategies in the Precision and Recall metrics. In general, the masking operation is effective in improving the model’s ability to capture relationships and contextual features among temporal sequences.

We also found it interesting that in the absence of graph masking operations, the initial graph structure input can significantly affect the training and convergence speed (requiring an additional 5+ epochs), despite our strategy to assist in the initial graph creation. However, after we performed the graph node masking operation, we were able to significantly reduce the convergence time of the training. That is, graph node masking enhances the model’s ability to construct graph structures and reduces the instability caused by node embedding when building the graph.

### 4.7. Hyperparameter Analysis

To evaluate the effect of different masking ratios on the model’s performance, we kept other hyperparameters fixed and adjusted the masking ratio of the time series and graph. We changed the input window of the model to 10 to achieve a masking ratio accuracy of 0.1 for the temporal sequences, meaning that MGUAD would randomly set the time at one moment of each sequence to 0. Note that the model may not score optimally on the metric due to the change in the timing window, but we can determine the trend through experiments using different sizes of the timing window. Given the proven necessity of the masking operation, we conducted experiments using masking ratios ranging from 0.1 to 0.6, and the results are shown in Figure 5.

In Figure 5, we show the performance of the model on the KDDCUP, SWaT, and WADI datasets using different masking ratios. The blue line represents Precision, the green line represents Recall, and the red indicator represents the F1_score. On the KDDCUP dataset, the model achieved a Recall close to 1.0 at a masking ratio of 0.1, which decreased significantly after the masking ratio exceeded 0.3. On the SWaT dataset, the model was not very sensitive to the masking ratio and performed more consistently. On the WADI dataset, MGUAD performed relatively well when the control masking rate was around 0.2. In summary, to efficiently complete the anomaly detection task, we can initially set the masking rate at 0.2, which then can be adjusted within a small range between 0.1 and 0.3 when the model is tested on different datasets.

## 5. Conclusions

Unsupervised multivariate time-series anomaly detection is particularly important in real-world applications. In this paper, we propose a novel multi-masking model, MGUAD, which can effectively capture the temporal and spatial correlations present in the input data. MGUAD can automatically build the graph structure of dependency relationships between sensors and update this structure as the relationships change. An ablation study has been conducted to verify the contribution of the multi-masking strategy in the proposed model, which enhances the robustness and learning abilities of the model and makes it easy to deal with difficult and unbalanced data. We have also conducted comprehensive experiments on three public datasets. The experimental results demonstrate that our model outperforms state-of-the-art anomaly detection methods. For future work, we intend to extend our experiments to optimize MGUAD in terms of building initialization graphs, the hyperparameters of the model, masking strategies, etc. We also aim to conduct further research on the interpretability of this GNN-based model.

## Figures and Tables

**Figure 1 sensors-23-07552-f001:**
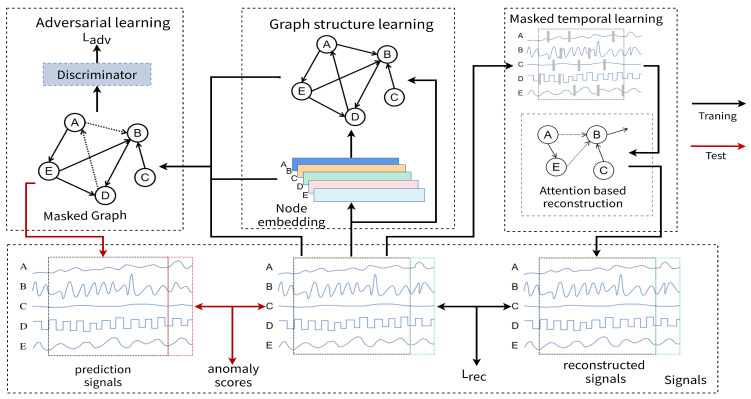
The proposed MGUAD architecture. A graph structure learning module is used to extract the spatial temporal dependencies between various sensors, and a masked temporal learning module is used to extract the correlation features between different timestamps. MGUAD uses a joint optimization network to generate the reconstruction and prediction values. In the training phase, we employ the reconstructed loss and the adversarial loss to optimize MGUAD. In the inference phase, the prediction values are further used to determine the anomaly scores for anomaly detection.

**Figure 2 sensors-23-07552-f002:**
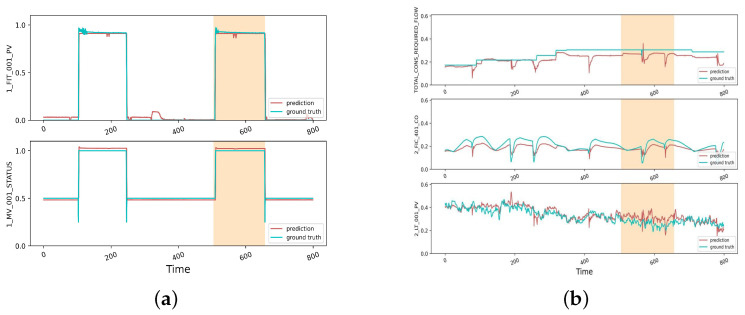
Detection data and prediction data of sensors during abnormal times. (**a**) Trends between two highly correlated sensors. (**b**) Nodes affected by anomalous attacks.

**Figure 3 sensors-23-07552-f003:**
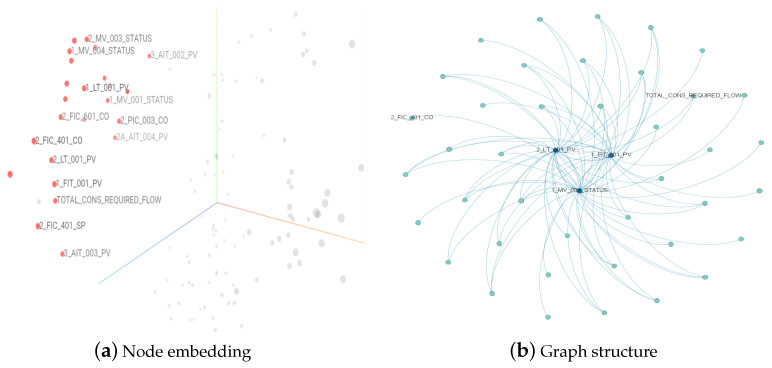
The structure of sensors in model learning. Sensors in close proximity in (**a**) exhibit more similar characteristics, and adjacent nodes in (**b**) show some correlation.

**Figure 4 sensors-23-07552-f004:**
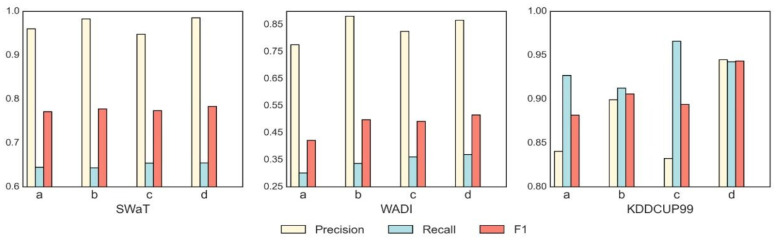
Performance comparison of four masking strategies on three datasets. From left to right, the performance on the SWaT, WADI, and KDDCUP99 datasets is shown. In each graph, a, b, c, and d represent the model without masking, with time-series masking, with node masking, and with multi-masking, respectively, where the red part in each graph indicates the F1_score and the yellow and blue parts represent Recall and Precision.

**Figure 5 sensors-23-07552-f005:**
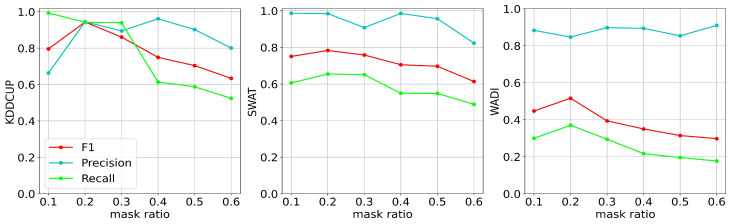
The performance of MGUAD under different masking ratios, where the horizontal coordinates represent the different masking ratios and the vertical coordinates represent the indicators corresponding to the model at that ratio.

**Table 1 sensors-23-07552-t001:** General information about the datasets.

Dataset	SWaT	WADI	KDDCUP99
Dimension	51	123	34
Training Size	16,097	9075	21,242
Testing Size	9700	6208	12,186
Train Rate	0.2772	0.2570	0.1799
Test Rate	0.0743	0.1369	0.5164

**Table 2 sensors-23-07552-t002:** Performance of MGUAD and 9 baselines on three datasets.

Dataset	Method	Precision	Recall	F1_score
SWAT	PCA	0.4712	0.4423	0.4563
	Isolation Forest	0.2201	0.6413	0.4270
	LSTM	0.5945	0.5276	0.5591
	DAGMM	0.7031	0.4713	0.5643
	LSTM_VAE	0.9540	0.5949	0.7328
	MAD-GAN	0.9333	0.6245	0.7483
	USAD	0.9635	0.6446	0.7724
	GDN	0.9499	0.6415	0.7658
	CAE	0.9845	0.5856	0.7343
	MGUAD	**0.9851**	**0.7204**	**0.8332**
WADI	PCA	0.3826	0.1822	0.2468
	Isolation Forest	0.1192	0.3425	0.1769
	LSTM	0.7242	0.2793	0.4043
	DAGMM	0.1691	**0.7869**	0.2783
	LSTM_VAE	0.4587	0.3212	0.3778
	MAD-GAN	0.1356	0.7273	0.2289
	USAD	0.8632	0.2787	0.3490
	GDN	0.8647	0.3508	0.5079
	CAE	0.4736	0.1652	0.2249
	MGUAD	**0.9281**	0.4114	**0.5701**
KDDCUP99	PCA	0.8544	0.3458	0.4923
	Isolation Forest	0.7896	0.9667	0.8692
	LSTM	0.7015	0.9038	0.7900
	DAGMM	0.8531	0.9746	0.9098
	LSTM_VAE	0.9285	0.8274	0.8751
	MAD-GAN	0.9628	0.7106	0.8178
	USAD	0.9569	0.9181	0.9370
	GDN	0.8494	0.9636	0.9029
	CAE	0.9241	0.5334	0.7288
	MGUAD	**0.9640**	**0.9818**	**0.9728**

**Table 3 sensors-23-07552-t003:** Average performance of our model and the baselines on all datasets.

		Precision	Recall	F1_score
	PCA	0.7360	0.3553	0.4792
	Isolation Forest	0.5994	0.9040	0.7209
	LSTM	0.6905	0.8339	0.7554
	DAGMM	0.7696	0.8980	0.8289
	LSTM_VAE	0.9210	0.7811	0.8453
	MAD-GAN	0.8231	0.6988	0.7559
	USAD	0.9567	0.8524	0.9071
	GDN	0.8587	0.9017	0.8797
	CAE	0.9258	0.5313	0.6751
	MGUAD	**0.9658**	**0.9339**	**0.9495**

**Table 4 sensors-23-07552-t004:** Performance of the three masking strategies.

Data	Mask	Precision	Recall	F1
SWAT	No mask	0.9598	0.6444	0.7712
	Time mask	0.9824	0.6435	0.7776
	Edge mask	0.9477	0.6540	0.7739
	Multi-mask	**0.9851**	**0.6547**	**0.7835**
WADI	No mask	0.7758	0.3015	0.4221
	Time mask	**0.8815**	0.3366	0.4872
	Edge mask	0.8253	0.3608	0.4923
	Multi-mask	0.8671	**0.3701**	**0.5162**
KDDCUP99	No mask	0.8405	0.9269	0.8816
	Time mask	0.8992	0.9125	0.9058
	Edge mask	0.8322	**0.9658**	0.8940
	Multi-mask	**0.9447**	0.9426	**0.9434**

## Data Availability

Data sharing does not apply to this article as no datasets were generated during the current study.

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
