# Peer review of "Masked Graph Neural Networks for Unsupervised Anomaly Detection in Multivariate Time Series"

_sensors, 2023, doi:10.3390/s23177552_

Round 1

Reviewer 1 Report

The paper proposes a novel model called MGUAD (Multi-Masking Graph-based Unsupervised Anomaly Detection) for unsupervised multivariate time series anomaly detection. MGUAD is designed to effectively capture temporal and spatial correlations in the input data, making it particularly suitable for real-world applications where anomalies may arise from complex interactions between different sensors or variables.

A) How does the multi-masking strategy in MGUAD contribute to enhancing the model's robustness and learning ability?  Explain how it handles difficult and unbalanced data?

B) Describe the process by which MGUAD automatically builds the graph structure of dependency relationships between sensors. How does the adaptive graph structure update as relationships change?

C) Enrich the literature by referring the recent works: A hypergraph based Kohonen map for detecting intrusions over cyber–physical systems traffic, Detection of false data cyber-attacks for the assessment of security in smart grid using deep learning.

C) In 4.7. Hyper-parameter Analysis, How did you handle hyperparameter optimization for MGUAD? Did you explore different hyperparameter tuning techniques, and what were the best settings you found?

D) In real-world applications, will the proposed MGUAD handle unseen anomalies or unexpected changes in the data distribution.

E) Analysis on graph structure and interpretability and anomaly detection performance may be considered.

The proposed MGUAD model seems to offer promising results in the domain of unsupervised multivariate time series anomaly detection. By addressing these suggestions, the research work on MGUAD can be further improved, making it more impactful and applicable to a wide range of real-world applications.

Minor editing of English language required

Author Response

Response to Reviewer 1 Comments

  1. A) How does the multi-masking strategy in MGUAD contribute to enhancing the model's robustness and learning ability? Explain how it handles difficult and unbalanced data?

Response: Thank you very much for your comments, and we have added the relevant descriptions in Sections 3.3 and 3.4. Our goal is to use the mask mechanism to enhance the robustness of autoregression-based anomaly detection. Through the mask operation, MGUAD can enhance the robustness and learning ability of the model in predicting the temporal data. MGUAD can forecast the future values in the absence of contextual information. By randomly masking multivariate sequences, the MGUAD can be forced to reconstruct each other through the context of each sequence and the correlation between different sequences. This operation can enhance the modeling ability of the model for difficult data. In real-world application scenarios, the multivariate time series data is often incomplete. Our model can use the incomplete data after a random mask as input, which makes it more capable of mining the complex information inside and between sequences. Our approach is unsupervised prediction-based approach. Only normal data needs to be utilized at the time of training to learn the representation of normal data. For data with unbalanced positive and negative samples, only the data of positive samples need to be utilized. And we demonstrated the effect of the mask operation via ablation experiments.

  1. Describe the process by which MGUAD automatically builds the graph structure of dependency relationships between sensors. How does the adaptive graph structure update as relationships change?

Response : Thank you very much for your comments. To construct the graph structure, each sequence is modeled as a node. The initial graph structure is obtained by computing the correlation between sequences. Then the graph structure will be updated by calculating the similarity between the node embedding. As the reviewer suggested, we have added some description in subsection 3.2.1. The graph structure is initialized before training, so that when the sensor relationship changed, the graph model is automatically reconstructed before training. When the sensors change, as described in subsection 3.2.2, the model will be updated by the similarity of the embedding. We have added a description of this process in subsection 3.2.

  1. C) Enrich the literature by referring the recent works: A hypergraph based Kohonen map for detecting intrusions over cyber–physical systems traffic, Detection of false data cyber-attacks for the assessment of security in smart grid using deep learning.

Response: Thanks for your advice, and we have been added these latest works.

  1. D) In 4.7. Hyper-parameter Analysis, How did you handle hyperparameter optimization for MGUAD? Did you explore different hyperparameter tuning techniques, and what were the best settings you found?

Response: Thanks for your advice. We explore some important hyperparameters, including learning rate, mask rate, and other hyperparameter settings in subsection 4.3, which we believe can support model replication or application.

  1. E) In real-world applications, will the proposed MGUAD handle unseen anomalies or unexpected changes in the data distribution.

Response: Thanks for your advice. Unseen anomalies with changed distributions are important components of anomalies. Our approach is essentially unsupervised and pattern-shifting techniques. MGUAD is a prediction-based model, so it does not learn the distribution of the anomalous data directly, but to learn the distribution of normal data. The unseen anomalies will be different from the normal data, so the MGUAD can use this deviation to detect the anomalies and be able to handle such unseen anomalies. We have also added explanations in section 2.1.3.

  1. F) Analysis on graph structure and interpretability and anomaly detection performance may be considered.

Response: Thanks for your advice. We can present the effect through the qualitative analysis experiment in Section 4.5. We explain the effect of the graph structure by showing the node embedding, and the correlation between different nodes. In our future research, we will further supplement and analyze the issues you raised, including the interpretability of graphs at the theoretical level.

Reviewer 2 Report

Shown in the attached file.

N.A.

Author Response

Response to Reviewer 2 Comments

Point 1. The abstract and introduction lack descriptions of the industrial background. All terms used in this study come from the field of computer science or mathematics. The authors have not assigned engineering meanings to them. For example, on line 5, what is "complex"among sensors; line 11, "hidden relationship”(better to used"unobserved causality""); line 34, what are "complexity" and "diversity" of data; line 46,"complex computation patterns"; line 52, is the"correlation" same to “causality"? line 54, what is "spatial information"; line 57, what kind of data lacks “clear graph topologies"; line 67, what is "graph-level context", and so on. These descriptions lack in-depth explanation from the perspective of industries. Some examples should be provided to clarify the engineering meanings of these terms. Otherwise, this manuscript may be suitable for a computer-oriented journal rather than engineering.

Response 1: Thank you very much for your meticulous reminder and examples. We will carefully revise each point you have pointed out in the abstract and introduction to add engineering meanings. 

  • "Complex associations" among sensors refer to the potentially intricate relationships between sensors, which might be positive or negative, direct or subtle, stable or subject to change over time.
  • "hidden relationship" → "unobserved causality"
  • "complexity" and "diversity" of data: “For example, anomalies where the absolute value of the deviation is not very large but the trend of the sequence is different.”
  • "complex computation patterns": The "complex computation patterns" is to emphasize that LSTMs involve intricate calculations, particularly due to the interplay of gates and memory cells.
  • Yes, the "correlation" can be exchanged to “causality".
  • "spatial information"→“ causality between sensors”
  • “clear graph topologies" → added “(the relationship between sensors is sometimes implicit)”
  • "graph-level context"→added “(e.g., interaction between different time series)”

We are committed to grounding our research applications in engineering.

Point 2. Some statements are arbitrary. For example, on line 37, what is "priori knowledge" when using PCA; on line 58, abbreviation GDN lacks full title, and on line 61, why GDN cannot consider temporal dependence? In my opinion, it is easy to consider it with GDN by using LSTM as based models. Also on line 128, GAN is just a framework, and we can consider correlation and temporal dependence using appropriate basic models. This should be explained in detail.

Response 2: Thanks for your advice. We apologize for our negligence in detail. PCA searches for the primary directions of variation, and then projects the data onto these principal components. However, the differences between normal and anomalous data might not solely manifest in the primary directions, but could involve other factors as well. "Priori knowledge" refers to domain-specific expertise, experience, or information that is known beforehand. In the absence of priori knowledge, PCA might be susceptible to the influence of random fluctuations or unrelated variations, potentially leading to false positives or missed detections.

The full title of GDN has been added to the text. As to the question of why GDN can't take into account temporal dependencies, the infrastructure of GDN doesn't use temporal networks such as LSTM. Hence, it lacks a certain ability to model temporal dependencies. Adding LSTM to the GDN's base module can improve the model's expressive ability, which is like the comparison method “LSTM-VAE”. We have verified in our experiments that our method has some enhancement to the LSTM based model.

We have added in the section 2.1.2 that GAN requires appropriate basic models to consider correlation and temporal dependence.

Point 3. For real application, it is better to design a method with simple structures and training strategies, unless the complex strategy and structure can contribute significantly to the results. The author should clarify the effectiveness of each strategy, including mask and adversarial training. The main contribution of this work is to detect anomaly considering data causality, so the GNN is necessary. However, it seems that the purpose of both mask and adversarial training is to improve the generalization and prevent overfitting. Their importance to the results needs to be discussed and analyzed in depth.

Response 3: Thanks for your advice. The mask operation on the edges is mainly to improve the robustness of the model. For real application, the obtained data are sometimes missing and incomplete, and the relationship between sensors is sometimes perturbed. By performing mask operations on edges, we can utilize less information to make predictions in the training phase, thus enhancing the learning ability of the model. In the inference phase, the model can make predictions with incomplete information in reality. Specifically, we add some descriptions in subsections 3.3 and 3.4. At the same time, we present the necessity of each component in the model through ablation experiments. In the future, we will try to streamline the model to simplify engineering operations.

Point 4. The inputs and outputs of each model and each part should be presented more clearly.

Response 4: Thanks for your advice. We apologize for our lack of clarity and have corrected it in sections 3.

Reviewer 3 Report

Acceptable but a couple of typos could be seen.

Author Response

Response to Reviewer 3 Comments

* How does the proposed architecture (Figure 1) leverage GNNs to achieve the specific goal as stated in the abstract, and what benefits do they provide?

Response: Thanks for your suggestion. By modeling sensors as graph nodes of the GNN and relationships between sensors as edges of the graph, the GNN is able to utilize its feature extraction capabilities. Gnn can exploit the features of among sensors, thus has greater ability of expression than traditional neural networks.

* What exactly is the supervised loss function (line 197)? How does it ensure the effective learning of anomalies?

Response: Thanks for your advice. In the training phase, by reconstructing the loss as described in Eq. (7), the model reconstructs the masked data to learn the distribution of normal data. In the inference phase, the model predicts the value based on the learned representations, and the value with a large deviation will be detected as anomalies. In other words, our model learns normal data instead of learning anomalies directly, using a prediction-based approach as described in Section 2.1.3. So that our model can handle unseen anomalies. We are sorry for our unclear expression, which is rectified in Section 3.1.

* Elaborate on how masking edges does relate to forecasting future values. How does the graph attention mechanism (line 198) contribute to forecasting accuracy?

Response: Thanks for your advice. The mask operation on the edges is mainly to improve the robustness of the model. In real engineering applications, the obtained data are often missing and incomplete, and the relationship between sensors is sometimes perturbed. By performing mask operations on edges, we can utilize less information to make predictions, thus enhancing the learning ability of the model. The edge masking is suitable for application scenarios with incomplete information in reality. The graph attention mechanism works by assigning different weights to different neighbors. That is, different neighbor nodes are considered to have different effects to simulate the strong and weak relationships between sensors in reality. The graph attention mechanism is more reasonable than the traditional GNNs’ information transfer method, so it can improve prediction accuracy.

* How does the discriminator (line 199) determine whether the predicted value is real or not? What criteria or features does it consider?

Response: Thanks for your advice. MGUAD uses transformer as discriminator, which needs to distinguish between the predicted data and the real time series data. That is, the discriminator will distinguish between two time series that differ only at the last time point, one of which is the true series and the other contains a predicted value. A detailed description is given in subsection 3.5, where Eq. (13) specifies the inputs to the discriminator and the formal expression of the loss.

* Line 504: What are index5 and index9?

Response: We are sorry that our version of the explanation did not match the picture. The coordinates in the picture are already the names of the sensors and do not need an index to indicate them. We have corrected this in the text.

* Define the masking ratios used in the paper.

Response: Thanks for your advice. Usually, the time series masking ratio is set to 20%. We give details of the experimental setup in subsection 4.3. And the experiments on mask rate in subsection 4.7 show that it is reasonable to fine-tune the mask rate between 10% and 30% for different datasets.

* Line 368: give some examples of “under assault in various circumstances”

Response: Thanks for your advice. The SWaT dataset is sensor data for water management detection. Modern Water Distribution Systems rely on computers, sensors, and actuators for both monitoring and operational purposes. The physical processes and embedded systems (cyber-physical systems) improve the level of service of water distribution networks but exposes them to the potential threats of cyberattacks. Specific malware attacks include historian data exfiltration attacks and process disruption attacks. That is, there may be a change in sensor detection data, a sudden pause in some normal supply, etc. We have added a related narrative in subsection 4.1.

* Illustrate with an example using the water security treatment systems (line 364) to show the applicability of the proposed Algorithm 1.

Response: Thanks for your advice. Since the water security treatment systems data is multivariate time series data, it can correspond to the inputs of Algorithm I. Then each time series is considered a node, and the nodes are embedded, after which the graph structure is built by calculating the similarity between the embeddings. Then the data is passed through multiple masks and fed into the generator to predict new data. The predicted data are fed it into the discriminator along with the original data. In the training phase, the embedding is updated with the model parameters via the final loss function. In the inference phase, the water resource data is fed, and then anomaly detection is done by calculating the deviation between predicted data and the original data.

* Line 533: How are masks added to the base model? Any algorithm used?

Response: Thanks for your advice. We stochastically mask the edges at a certain rate, and we add related explanation in Section 4.6.

Round 2

Reviewer 1 Report

The authors have presented Masked Graph Neural Networks for Unsupervised Anomaly Detection in Multivariate Time Series. The paper is generally well-written and structured, I am fine with the content and no further changes are required from my side.

Reviewer 2 Report

All comments were well modified and responded to. The paper can be accepted.

Reviewer 3 Report

The concerns have been resolved.